# INSTRUCTION-FREE TUNING OF LARGE VISION LANGUAGE MODELS FOR MEDICAL INSTRUCTION FOLLOWING

## ABSTRACT

Large vision language models (LVLMs) have demonstrated impressive performance across various tasks, but struggle in domains with limited data, such as medicine. While visual instruction tuning addresses this by fine-tuning models with instruction-image-output triplets, constructing large-scale and high-quality datasets remains challenging in domains requiring expert knowledge. To address this, we introduce an instruction-free tuning that reduces reliance on handcrafted or auto-generated instructions, leveraging only image-output pairs during fine-tuning. Specifically, we propose a momentum proxy instruction as a replacement for explicit instructions, preserving the instruction-following capability of the pre-trained LVLM while promoting refined updates for parameters that remain valid during inference. Consequently, the fine-tuned LVLM can flexibly respond to domain-specific instructions, even when explicit instructions are absent during fine-tuning. Additionally, we incorporate a response shuffling strategy to mitigate the model's over-reliance on previous words, facilitating more effective fine-tuning. Our approach achieves state-of-the-art accuracy on multiple-choice visual question answering tasks across SKINCON, WBCAtt, and CBIS datasets, significantly enhancing fine-tuning efficiency in medical domains.

## 1 INTRODUCTION

Large vision language models (LVLMs) have demonstrated general-purpose capabilities across various language understanding and generation tasks (Huang et al., 2023). Despite this success, their performance often declines in domains with limited publicly available data, such as medicine (Li et al., 2023). Visual instruction tuning (Liu et al., 2023), which fine-tunes LVLMs on image-instruction-output triplets (as illustrated in Fig. 1(a)), has been introduced to enhance adherence to domain-specific instructions and has achieved notable success.

However, the performance of fine-tuned models is highly dependent on the scale and quality of the instruction dataset (Wei et al., 2022; Zhou et al., 2023), incurring substantial costs in dataset construction. Therefore, various methods (Shengyu et al., 2023) have leveraged large language models (LLMs) to automatically construct instruction datasets, thereby reducing human involvement. These attempts have also been extended to the medical domain (Li et al., 2023), where raw data paired with images (e.g., radiology reports or captions) serves as a valuable resource for constructing instruction datasets. However, the frequent use of domain-specific abbreviations and terms (Moon et al., 2014), along with the need for expert-level contextual understanding (Leaman et al., 2015), poses a significant challenge for constructing datasets using LLMs. Furthermore, radiology reports typically begin with a lesion description and conclude with a diagnostic impression (Kahn Jr et al., 2009); the stronger semantic correlation between sentences, relative to natural image texts, tends to exacerbate hallucinations (Favero et al., 2024). Therefore, we revisit fine-tuning approaches that reduce reliance on handcrafted or auto-generated instructions (as illustrated in Fig. 1(b)), aiming to avoid inefficiencies in constructing instruction datasets and to mitigate potential drawbacks caused by inadequate instructions.

*Instruction-free tuning* refers to fine-tuning LVLMs using only image-output pairs, without relying on any instructions. To the best of our knowledge, this is the first work to formalize and address

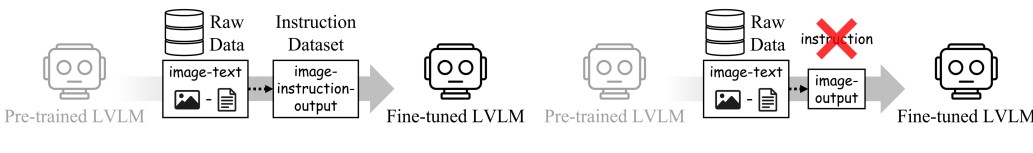

(a) Instruction Tuning                                  (b) Instruction-Free Tuning

Figure 1: Conceptual illustrations of (a) instruction tuning, which fine-tunes the model on curated instructions and outputs (e.g., *"Why does the image..."* and *"Because it shows..."*), and (b) instruction-free tuning, which fine-tunes the model solely on paired textual descriptions (e.g., radiology reports). Instruction tuning requires instruction-image-output triplets constructed by humans or LLMs prior to fine-tuning, whereas instruction-free tuning can be performed on image-output pairs without additional steps.

this task. However, fine-tuning a model without instructions introduces an additional layer of challenges, especially when it exploits the inherent properties of paired raw data as instructions (e.g., *"Describe this medical scan"*) (Sellergren et al., 2025). Due to the characteristics of medical data, the paired radiology report does not perfectly align with the natural response (e.g., *"The image..."*) that would typically follow a conversational instruction (e.g., *"Describe..."*). The misalignment between the instruction-output pairs in the fine-tuning and pre-training datasets poses a risk of degrading the instruction-following capability of the pre-trained LVLM. Specifically, this may lead to *image-dependence*, where the model loses its pre-trained instruction-following capability and generates fixed-style responses that rely excessively on the image regardless of the given instruction (see Fig. 6). Alternatively, this may result in *instruction-dependence* (or *parameter-dependence*), where the model responds appropriately only to instructions encountered during fine-tuning and fails to generate responses aligned with the fine-tuning data for unseen instructions (instead respond based on the pre-trained knowledge; see other LVLMs in Table 1). Furthermore, due to the nature of instruction-following LVLMs, instructions seen during fine-tuning are not guaranteed to be provided at inference time, highlighting the importance of preventing overfitting to such instructions.

In response to these challenges, we introduce a *proxy instruction* that is optimized to align with the set of image-output pairs in the fine-tuning dataset. This enables the LVLM to adapt to domain-specific datasets without degrading its instruction-following capability by employing a well-aligned instruction. As a result, the fine-tuned model can flexibly respond to a wide range of text instructions in the medical context during inference. Furthermore, we refined the proxy instruction through an exponential moving average, i.e., a *momentum proxy instruction*. This mitigates overfitting to a specific proxy instruction and allows the remaining parameters to learn drift-compensated representations that reflect overall fine-tuning trends. Meanwhile, fine-tuning a model while maintaining its pre-trained instruction-following capability can mitigate the challenges posed by the unique characteristics of medical data. Since the diagnostic conclusion can often be inferred from the earlier words (e.g., findings) alone, the model is likely to largely overlook visual features when generating the latter part of the response. In this context, we propose a strategy, *response shuffling*, which randomly shuffles the sentence order of the ground truth output, thereby preventing the model from relying too heavily on previous words. Notably, employing a proxy instruction disregards the negative effects of shuffling output order during fine-tuning, enabling a simple strategy to substantially enhance medical visual question answering (VQA) performance.

In summary, the contributions of this paper are as follows:

- We introduce and formalize *instruction-free tuning*, a novel paradigm for fine-tuning LVLMs that eliminates the need for handcrafted or auto-generated instructions, addressing a key bottleneck in adapting models to specialized domains.

- We propose a *momentum proxy instruction* designed to preserve the instruction-following capability of the pre-trained LVLM during fine-tuning, while promoting refined updates for parameters that remain valid during inference. Additionally, we propose a *response shuffling* strategy to mitigate the model's over-reliance on previous words, facilitating more effective fine-tuning of medical LVLMs.

- We demonstrate that our fine-tuned model achieves state-of-the-art accuracy in multiple-choice VQA evaluations on the SKINCON, WBCAtt, and CBIS datasets.

## 2 RELATED WORKS

**Instruction Tuning.** Instruction tuning fine-tunes a model using a dataset consisting of instruction-input-output triplets. In particular, the instruction specifies the task (i.e., a question), the input provides supplementary context (e.g., an image), and the output is the expected response based on the instruction and input. As one of the early studies, Natural Instructions (Mishra et al., 2022) fine-tuned a model with an instruction dataset constructed by integrating 61 datasets. Recently, an alternative approach that constructs instruction datasets using LLMs was proposed; Alpaca (Taori et al., 2023) efficiently fine-tuned the LLaMA (Touvron et al., 2023) using 52K instruction-output pairs generated by an instruction-tuned GPT-3.5 (Achiam et al., 2023). Moreover, Self-Instruct (Wang et al., 2023) proposed a method for enabling an LLM to follow diverse instructions by iteratively generating and fine-tuning on synthetic instruction-output data with minimal human intervention. In parallel, one study showed that instruction tuning remained effective even when instructions were replaced with an empty string during fine-tuning (Hewitt et al., 2024). However, these methods are incapable of handling scenarios involving visual data and are less effective at improving the performance of fine-tuned models when LLMs possess limited domain-specific knowledge.

**Visual Instruction Tuning.** Building on the success of instruction tuning, visual instruction tuning with LLMs was introduced for vision tasks. LLaVA (Liu et al., 2023) integrated CLIP's vision encoder (Radford et al., 2021) with LLaMA (Touvron et al., 2023), and was fine-tuned on an instruction dataset generated by language-only GPT-4 (Achiam et al., 2023). Recently, LLaMA-3.2-Vision (Dubey et al., 2024) extended LLaMA 3.1 by adapting cross-attention layers to integrate the image modality with language. Additionally, alternative LLMs have been integrated into the vision domain, such as Qwen2.5-VL (Bai et al., 2025), which utilizes patch grouping and window attention to efficiently process large numbers of visual tokens. These successes in LVLMs extended to medical domains. LLaVA-Med (Li et al., 2023) was fine-tuned on an instruction dataset generated by GPT-4 (Achiam et al., 2023), based on biomedical image-caption pairs from PubMed Central (PMC). Recently, PubMedVision (Chen et al., 2024) and MedGemma (Sellergren et al., 2025) have been fine-tuned on carefully curated medical datasets from PMC and other publicly available sources, significantly enhancing their medical capabilities. One study (Anonymous, 2025) attempted instruction-free tuning by using language-only instruction-output pairs and image-caption pairs with a fixed set of instructions. However, none of the existing methods completely avoided employing text instructions during LVLM fine-tuning. To our knowledge, this is the first approach that enables fine-tuning of LVLMs without using any text instructions.

## 3 METHOD

### 3.1 INSTRUCTION-FREE TUNING

We start by formulating instruction-free tuning as follows: given an image $X_v$, a prompt $X_p$ containing an instruction, and the ground truth output $y$, instruction-free tuning fine-tunes a model to follow diverse instructions using $(X_v, y)$ pairs rather than (instruction, $X_v$, $y$) triplets. However, the model fine-tuned on $(X_v, y)$ using raw data properties as instructions (e.g., *"Describe this medical scan"*) may lead (a) *image-dependence*, generating fixed-style responses regardless of the instruction; or (b) *instruction-dependence*, responding well only to instructions identical to those seen during fine-tuning. To address these issues, we introduce a *proxy instruction*, which is well-aligned with $y$ and is further refined using an exponential moving average, i.e., a *momentum proxy instruction*. This approach preserves the pre-trained instruction-following capability and promotes refined updates for parameters that remain valid during inference (i.e., the vision encoder).

**Proxy Instruction.** Our framework is illustrated in Fig. 2. It comprises a vision encoder $g$ (a vision transformer (Dosovitskiy et al., 2021) followed by projection layers) and a language model $f$. To generate responses from both the image and the instruction, $X_v$ is fed into $g$ to extract the key-value matrices $KV_v$. Then, $f$ generates a response $\hat{y}$ by integrating the query matrix of $X_p$ with $KV_v$ through its cross-attention layer (Dubey et al., 2024; Alayrac et al., 2022). Let the proxy instruction $t = \{t_1, \ldots, t_N\}$ be a set of $N$ continuous vectors, each with the same dimension as the word embeddings. We feed $t$ into $f$ as input and use it to replace the text instruction in $X_p$. Since $f$ is differentiable, gradients can be propagated to update $t$, enabling the model to derive an optimal instruction from the data. For optimizing $t$ (and the learnable/frozen parameters of $g$), we compute

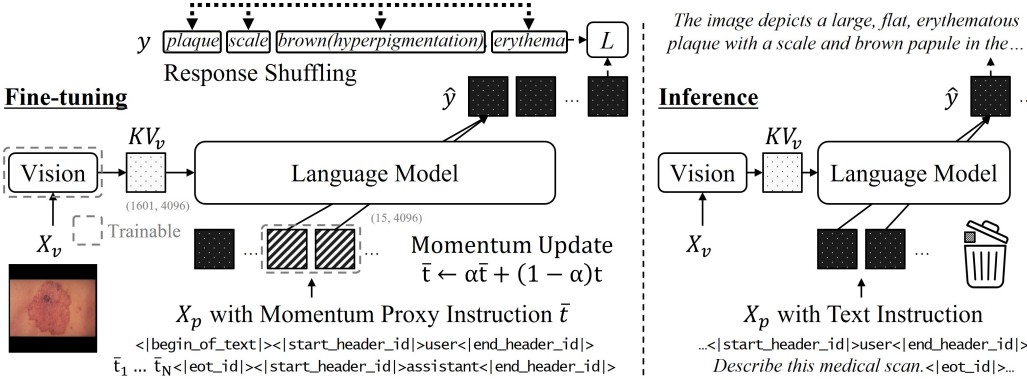

Figure 2: An illustration of our instruction-free tuning framework. The vision encoder extracts key-value matrices $KV_v$ from an image $X_v$, which are then integrated into a prompt $X_p$ for the language model to generate the response $\hat{y}$. For instruction-free tuning, the text instruction in $X_p$ is replaced with the momentum proxy instruction $\bar{t}$. During supervised fine-tuning, $g$ is updated with autoregressive loss $L$ between the response $\hat{y}$ and the ground truth $y$. In parallel, (warm-up initialized) $\bar{t}$ is gradually updated via exponential moving average of the proxy instruction $t$. During inference, $\bar{t}$ is discarded, and a conversational text instruction (e.g., *"Describe..."*) is used to generate a natural language response (e.g., *"The image depicts..."*).

an autoregressive loss $L$ between $y$ and $\hat{y}$, while keeping the parameters of $f$ frozen, following (torchtune, 2024). Formally, supervised fine-tuning is defined as:

$$L(g, t) = -\sum_{i \in \mathcal{I}_Y} \log f(y_i \mid H_{<i}(X_v, X_p; g, t)), \tag{1}$$

where $\mathcal{I}_Y$ denotes the indices corresponding to the output sequence $y$, and $H_{<i}$ denotes the past activations in the left context of $X_v$ and $X_p$. For simplicity, supplementary tokens in $X_p$, such as `<|eot_id|>`, are omitted. Notably, the proxy instruction differs from that used in Prefix-Tuning (Li & Liang, 2021). In Prefix-Tuning, the learnable vectors are used during inference, whereas in our method they serve as a proxy during fine-tuning and are discarded at inference.

**Momentum Proxy Instruction.** Although we obtain an optimized instruction $t$ that is well-aligned with its corresponding set of outputs, $t$ becomes outdated during fine-tuning and is replaced with a conversational text instruction at inference. In response to this, we gradually update $t$ using an exponential moving average to capture overall fine-tuning trends (He et al., 2020), thereby promoting refined updates to $g$, which remains valid during inference. Formally, we update the momentum proxy instruction $\bar{t}$ as follows:

$$\bar{t} \leftarrow \alpha \bar{t} + (1 - \alpha)t, \tag{2}$$

where $\alpha \in [0, 1)$ denotes the momentum coefficient. The parameters $g$ are updated immediately via Eq. 1, whereas $\bar{t}$ evolves gradually. The overall instruction-free tuning procedure with the momentum proxy instruction is summarized in Appendix (see Algorithm A.1). Note that $\bar{t}$ is initialized in a warm-up stage by optimizing $t$ with $g$ kept frozen.

## 3.2 RESPONSE SHUFFLING

In medical domains, the activations of $X_v$ in $H_{<i}$ are likely to be largely overlooked during supervised fine-tuning. To address this, we randomly shuffle $y$ during fine-tuning to prevent the model from relying on the previous word when generating a response. Formally,

$$\text{RS}(y) = \text{Join}(\text{Shuffle}(\text{Split}(y))), \tag{3}$$

where `Split` segments $y$ by a separator (e.g., ","), `Shuffle` randomly permutes the segments, and `Join` reconstructs the permuted $y$ using the original separator. Notably, the goal of our fine-tuning is not to train the model to generate high-quality radiology reports, but rather to adapt the model to a domain-specific dataset while preserving the pre-trained instruction-following capability. Therefore, the negative effects of shuffling the output order during fine-tuning are disregarded (see Fig. 6).

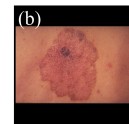

(a) big cell size, round cell shape, unsegmented-band nucleus shape, low nuclear cytoplasmic ratio, densely chromatin density, clear cytoplasm texture, light blue cytoplasm colour, granularity, small granule type, pink granule colour, Neutrophil

(b) plaque, scale, brown(hyperpigmentation), erythema

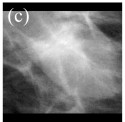

(c) oval mass shape, circumscribed mass margins, mass, benign

Figure 3: Examples of the medical report for the (a) SKINCON, (b) WBCAtt, and (c) CBIS datasets.

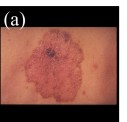

(a) What is the dermatologic feature?

A. nodule, crust, dome-shaped, erythema
B. nodule, exophytic/fungating
C. plaque, scale, brown(hyperpigmentation), erythema
D. papule, purpura/petechiae, black
E. plaque, atrophy, brown(hyperpigmentation)
…answer is [A or B or C or D or E]."

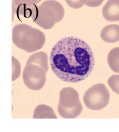

(b) What is the cell size?

A. big  B. small

Response with only the letter of the correct choice, starting with "The correct answer is [A or B]."

Figure 4: Examples of multiple-choice VQA for the (a) SKINCON and (b) WBCAtt datasets.

## 4 EXPERIMENTS

**Datasets.** We conducted experiments on three richly annotated medical datasets, as shown in the Fig. 3. **SKINCON** (Daneshjou et al., 2022) is a dermatology dataset that contains 3230 images from the Fitzpatrick17k (Groh et al., 2021). Each image was manually annotated by dermatologists using 48 clinical attributes, such as *plaque*. The dataset was split into 80% for training, 10% for validation, and 10% for testing. **WBCAtt** (Tsutsui et al., 2023) is a dataset consisting of 10298 microscopic images of white blood cells, such as *neutrophils*. Each image was annotated with 11 attributes, such as *cell size*, based on guidelines defined by pathologists. The dataset was split into 6179 images for training, 1030 for validation, and 3099 for testing. **CBIS** (Lee et al., 2017) is a digitized mammography dataset consisting of 892 mass cases and 753 calcification cases. Each mammogram was annotated with *pathology*, *abnormality types*, and BI-RADS descriptors (e.g., *mass shape*). This dataset provides cropped images of lesions extracted through ROI segmentation. For our experiments, we used cropped images along with their corresponding attribute annotations. The dataset was split into 90% for training and 10% for validation, and the official test split was used for testing.

**Multiple-choice VQA.** To evaluate the fine-tuned LVLMs in instruction-free tuning scenarios, we converted the training split data into a medical report (see Fig. 3), while the validation and test split data into a multiple-choice VQA (see Fig. 4). When converting to medical reports, attribute annotations were formatted into plain text. In the SKINCON dataset, a medical report was formatted by listing the names of the items marked as *present* among the 48 attributes, such as *"plaque, scale."* In the WBCAtt dataset, a medical report was formatted by listing 8 attributes in the form of {*value*} {*name*} (e.g., *"big cell size"*). Meanwhile, yes/no attributes, such as *granularity*, were added using the {*name*} when present, whereas *label* attributes were added using the {*value*}. In the CBIS dataset, a medical report was formatted by listing the {*value*} of *abnormality* and *pathology* attributes. The remaining attributes were added in the form of {*value*} {*name*}. When converting to multiple-choice VQA, attribute annotations were formatted into question-answer pairs, usually with five options. The question was constructed using the attribute name in the form *"What is the {name}?"*, whereas for the SKINCON dataset, the fixed question *"What is the dermatologic feature?"* was used. For multiple-choice options, the annotated attribute value was set as the correct answer, while randomly sampled non-overlapping values from the same attribute served as the incorrect options. Attributes with fewer than five options (e.g., yes/no) were converted into a limited choice set (e.g., two options). At the end, the response style was specified, such as *"Response with…"* (see Fig. 4), and the predicted choice (e.g., A) was extracted by parsing the model's generated response. We evaluated multiple-choice VQA using accuracy by comparing the selected choices with the correct answers.

**Implementation Details.** The proposed method was developed on top of LLaMA-3.2-11B-Vision-Instruct (Dubey et al., 2024), and fine-tuning was performed using the torchtune framework (torchtune, 2024). For models with proxy instructions, the number of instruction vectors $N$ was set to 8, and the momentum coefficient $\alpha$ was set to 0.999. The warm-up stage was conducted for one epoch (approximately 3000 steps), while the fine-tuning stage was conducted for five epochs. We mainly followed the default settings in torchtune for supervised fine-tuning. We conducted each experiment three times with different seeds and report the average accuracy.

Table 1: Multiple-choice VQA accuracy on the SKINCON (SKN.), WBCAtt (WBC.), and CBIS datasets. LLaMA-3.2 denotes LLaMA-3.2-11B-Vision-Instruct (Dubey et al., 2024), Qwen denotes Qwen2.5-VL-3B-Instruct (Bai et al., 2025), PubMed denotes PubMedVision-7B-Qwen2.5VL (Chen et al., 2024), and MedGemma denotes MedGemma-4B-it (Sellergren et al., 2025). FT denotes fine-tuning, and RS denotes response shuffling. **Bold** indicates the highest accuracy, and underline denotes the second highest.

| Model | SKN. | WBC. | CBIS | Avg. |
|---|---|---|---|---|
| MedGemma (*w/o* FT) | 14.3 | 37.3 | 39.2 | 30.2 |
| PubMed (*w/o* FT) | 47.0 | 36.3 | 40.6 | 41.3 |
| Qwen (*w/o* FT) | 37.7 | 32.8 | 36.0 | 35.5 |
| LLaMA-3.2 (*w/o* FT) | 39.9 | 34.6 | 43.0 | 39.1 |
| MedGemma (FT) | 10.6 | 47.3 | 44.4 | 34.1 |
| Qwen (FT) | 42.4 | 39.7 | 39.4 | 40.5 |
| LLaMA-3.2 (FT) | 59.4 | 61.7 | 63.9 | 61.6 |
| **InstFree** | 69.8 | **68.4** | 65.0 | 67.7 |
| **InstFree *w/* RS** | **75.2** | 65.2 | **68.3** | **69.5** |

Table 2: Multiple-choice VQA accuracy of instruction-free tuning variants. Inst. denotes InstFree, while Fix and Update denote models fine-tuned with fixed and continuously updating proxy instructions, respectively.

| Method | SKN. | WBC. | CBIS | Avg. |
|---|---|---|---|---|
| Inst. *w/* Fix | 63.9 | 65.1 | 61.7 | 63.5 |
| Inst. *w/* Update | 66.4 | 63.6 | 63.7 | 64.5 |

Table 3: Ablation study on the response shuffling. Bal denotes balanced sampling; RS ("") and Rand denote models fine-tuned with an incorrect separator and randomized text instructions, respectively.

| Method | SKN. | WBC. | CBIS | Avg. |
|---|---|---|---|---|
| Inst. *w/* Bal | 65.4 | 63.4 | 60.8 | 63.2 |
| Inst. *w/* RS ("") | 67.9 | 57.9 | 67.2 | 64.3 |
| FT *w/* RS | 74.1 | 62.0 | 67.3 | 67.8 |
| FT *w/* Rand | 3.1 | 27.5 | 27.9 | 19.5 |

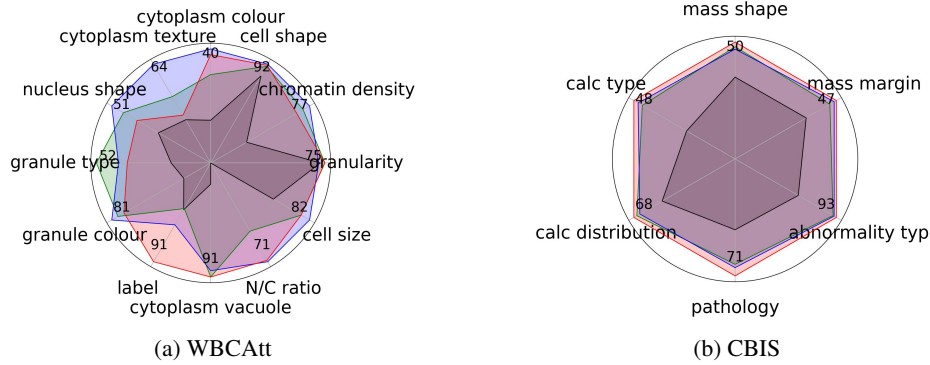

(a) WBCAtt    (b) CBIS

Figure 5: Radar charts of the accuracy for each attribute on the (a) WBCAtt and (b) CBIS datasets. LLaMA-3.2 (*w/o* FT) is shown in black, LLaMA-3.2 (FT) in green, InstFree in blue, and InstFree *w/* RS in red.

## 5 RESULTS

### 5.1 MULTIPLE-CHOICE VQA

**Main Results.** We first compared our method against other LVLMs without fine-tuning (*w/o* FT) on the SKINCON, WBCAtt, and CBIS datasets, including general LVLMs such as LLaMA-3.2-11B-Vision-Instruct (Dubey et al., 2024) (LLaMA-3.2) and Qwen2.5-VL-3B-Instruct (Bai et al., 2025) (Qwen), as well as medical LVLMs such as PubMedVision-7B-Qwen2.5VL (Chen et al., 2024) (PubMed) and MedGemma-4B-it (Sellergren et al., 2025) (MedGemma). We then compared our method with fine-tuned (FT) LLaMA-3.2, Qwen, and MedGemma using a text instruction (i.e., *"Describe this medical scan"*), with Qwen and MedGemma followed the fine-tuning settings described in their original papers. We refer our method as **InstFree**, which fine-tuned the model using the momentum proxy instruction, while **InstFree *w/* RS** fine-tuned it with both the momentum proxy instruction and response shuffling. Since the WBCAtt and CBIS datasets provided attribute names, we reported the accuracy for each attribute separately.

Table 1 shows the multiple-choice VQA accuracy on the SKINCON, WBCAtt, and CBIS datasets. The accuracy of models without fine-tuning (*w/o* FT) is consistently lower than that of fine-tuned models. Notably, MedGemma underperforms compared to non-medical LVLMs, which we at-

tribute to its primary fine-tuning on pathology data, suggesting that recent medical LVLMs may face challenges in generalizing across all types of medical data. Overall, these results highlight that fine-tuning LVLMs is crucial for downstream tasks in the medical domain, as it yields substantial performance improvements.

Although LLaMA-3.2 (FT) achieves comparable accuracy on the CBIS dataset, its average accuracy across all three datasets is lower than that of the proposed method, highlighting the limitations of naively incorporating a text instruction during fine-tuning. In contrast, the accuracies of Qwen (FT) and MedGemma (FT) are significantly lower than that of LLaMA-3.2 (FT). This discrepancy is not due to the limited model capacity of Qwen and MedGemma, but rather to their strong dependence on instructions, i.e., *instruction-dependence*. Specifically, the fine-tuned model generates correct responses to certain instructions (i.e., *"Describe this medical scan"*), but fails to respond appropriately to other types of instructions, such as *"What is the cell shape?",* instead responding based on its inherent pre-trained knowledge (Goyal et al., 2025). Notably, even though MedGemma has been fine-tuned, it still shows low accuracy on the SKINCON dataset. This is likely due to the absence of explicit attribute names in the medical reports and the limited instruction-following capability of the pre-trained LVLM, which further exacerbates the *instruction-dependence* of the fine-tuned model. In summary, the limited ability of fine-tuned Qwen and MedGemma to generalize across diverse instructions leads to a significant degradation in multiple-choice VQA performance.

Overall, InstFree achieves the highest accuracy across all datasets among the LVLMs compared, demonstrating the superiority of the momentum proxy instruction. In addition, employing response shuffling (i.e., InstFree *w/* RS) further improves the average accuracy to 69.5%, demonstrating the effectiveness of the response shuffling. Although it shows a decrease in accuracy on WBCAtt compared to InstFree, the advantages of our response shuffling are demonstrated in attribute-wise accuracy. Figure 5 shows radar charts of each attribute for the WBCAtt and CBIS datasets. While leveraging inter-attribute correlations can improve overall accuracy, over-reliance on them may compromise the reliability of certain attribute predictions. Specifically, the WBCAtt dataset is susceptible to misclassification since some labels can be inferred from earlier parts of a response (Tsutsui et al., 2023). In this context, the superior accuracy achieved by response shuffling for primary targets that have strong correlations (e.g., label) highlights the effectiveness of the proposed method in reducing over-reliance on previous words. Similarly, applying response shuffling improves the overall accuracy on both the CBIS and SKINCON datasets, demonstrating our method's effectiveness in enhancing the reliability of each attribute prediction across datasets.

**Comparison with Instruction-Free Tuning Variants.** We compared our method with two instruction-free tuning variants, i.e., Fix and Update, to demonstrate the effectiveness of the proposed momentum proxy instruction. Fix used a fixed proxy instruction derived from a prior warm-up stage and updated only the vision encoder during fine-tuning, whereas Update referred to fine-tuning in which both the proxy instruction and the vision encoder were updated via gradient descent.

Table 2 shows the multiple-choice VQA accuracy of instruction-free tuning variants. Among the variants, accuracy differs between the Fix and Update, as Fix achieves higher performance on WBCAtt, while Update shows higher accuracy on SKINCON and CBIS. The superior accuracy of WBCAtt with Fix might be due to the vision encoder having fewer changes during fine-tuning, probably because the cells are always centered, which leads to more consistent image features. Notably, employing the momentum proxy instruction outperforms Fix by more than 4%, confirming the benefits of capturing overall fine-tuning trends while prioritizing updates to the vision encoder parameters.

## 5.2 Ablation Studies

**Ablation on Response Shuffling.** We conducted ablation studies on response shuffling using the SKINCON, WBCAtt, and CBIS datasets. First, we compared InstFree w/ Bal (balanced sampling) to investigate whether the issue originates from the model overfitting to previous word correlations or recurring response patterns. We calculate the frequency of each word and assign higher selection probabilities to samples that contain words with lower overall frequencies. Second, we compared InstFree *w/* RS (" ") to demonstrate that accuracy degrades when response shuffling uses incorrect separators such as " " (instead of the ";" separator). Third, we compared FT *w/* RS to show that response shuffling can improve VQA accuracy even when LLaMA-3.2 is fine-tuned with a text instruction. Lastly, we compared FT *w/* Rand (fine-tuning LLaMA-3.2 with randomized text in-

structions) to demonstrate that diversifying instructions does not lead to better performance (as an extension of response shuffling). To do this, we randomly select an instruction from a set of 58 predefined text instructions, such as *"Please report this medical scan."* (see Appendix B).

Table 3 shows the accuracy of the ablation study on response shuffling across the SKINCON, WB-CAtt, and CBIS datasets. InstFree *w/* Bal shows lower accuracy compared to the model without balanced sampling, i.e., InstFree. These results indicate that oversampling of specific samples leads to overfitting and results in lower accuracy, suggesting that reducing over-reliance on previous words is more important. InstFree *w/* RS (" ") shows lower accuracy than the model that uses a comma "," as the separator. Although standard sentence segmentation libraries are generally sufficient, domain-specific datasets often contain text with poorly structured sentences; therefore, carefully designing strategies to separate elements is necessary for effective response shuffling. While FT *w/* RS shows higher accuracy than the model without response shuffling, its accuracy remains lower than that of InstFree *w/* RS, highlighting the benefits of the momentum proxy instruction. Meanwhile, FT *w/* Rand exhibits significantly lower accuracy. This suggests that the fine-tuned model tends to ignore the given instructions, consistently generating responses that follow the format of our medical reports (i.e., *image-dependence*), such as *"plaque, scale..."* This result implies that diversifying instructions during fine-tuning degrades the LVLM's instruction-following capability, thereby leading the model to generate responses primarily based on the visual input.

*Discussion of Misalignment.* Fine-tuning the model to generate consistent responses across a broad range of instructions leads to a significant loss of its pre-trained instruction-following capability (see Table 3). This appears to stem from a misalignment with the fine-tuning dataset, as the pre-training paired outputs vary in format depending on the instruction. Similarly, both the drift from the fixed proxy instruction during fine-tuning (see Table 2) and the misalignment between a naively paired single instruction and the pre-training dataset (see FT in Table 1) can be interpreted as factors that adversely affect the instruction-following capability. Meanwhile, continuously minimizing misalignment during fine-tuning may appear ideal; however, excessive reliance on instructions (which will be discarded) through direct gradient descent can ultimately lead to a degradation in VQA performance (see Table 2). Within this context of misalignment, since most publicly available LVLMs do not release their datasets, manually replicating such datasets to reduce misalignment is impractical. Therefore, momentum proxy instruction offers a practical solution for fine-tuning LVLMs without instructions.

**Ablation on Coefficients and Instruction Scale.** We conducted ablation studies on the momentum proxy instruction using the SKINCON, WBCAtt, and CBIS datasets by varying the momentum coefficient and the number of instruction vectors to identify optimal hyperparameters. For the momentum coefficient $\alpha$, we experimented with values of 0.9, 0.99, 0.999, and 0.9999, while setting the number of instruction vectors $N$ to 8. For the number of instruction vectors $N$, we experimented with values of 1, 2, 4, 8, 16, and 32, while setting the momentum coefficient $\alpha$ to 0.999.

Table 4 shows the accuracy of the ablation study on the momentum coefficient. The value 0.999 yielded the highest accuracy, indicating that excessively fast or slow adaptation may limit performance improvements, similar to (He et al., 2020). Table 5 shows the accuracy of the ablation study on the number of instruction vectors. The highest accuracy was achieved with 8 instruction vectors, whereas increasing the number to 32 resulted in lower performance. This may be related to general questions averaging about 8 tokens in length.

Table 4: Ablation study on the momentum coefficient $\alpha$.

| $\alpha$ | SKN. | WBC. | CBIS | Avg. |
|---|---|---|---|---|
| 0.9 | 64.1 | 64.2 | 60.7 | 63.0 |
| 0.99 | 68.0 | 64.0 | 60.9 | 64.3 |
| 0.999 | **69.8** | **68.4** | **65.0** | **67.7** |
| 0.9999 | 66.5 | 64.5 | 64.0 | 65.0 |

Table 5: Ablation study on the number of instruction vectors $N$.

| $N$ | SKN. | WBC. | CBIS | Avg. |
|---|---|---|---|---|
| 1 | 66.4 | 63.7 | 63.6 | 64.5 |
| 2 | 65.4 | 64.5 | 63.6 | 64.5 |
| 4 | 65.3 | 65.5 | 63.0 | 64.6 |
| 8 | **69.8** | **68.4** | **65.0** | **67.7** |
| 16 | 68.2 | 64.5 | 63.1 | 65.2 |
| 32 | 66.4 | 64.4 | 64.2 | 65.0 |

### 5.3 QUALITATIVE VQA ANALYSIS

Fig. 6 presents VQA results of LLaMA-3.2 on the WBCAtt dataset. In the case of descriptive instruction (a), the model without fine-tuning (*w/o* FT) identifies the image as a neutrophil, whereas fine-tuned models identify it as a monocyte, demonstrating that fine-tuning allows the model to iden-

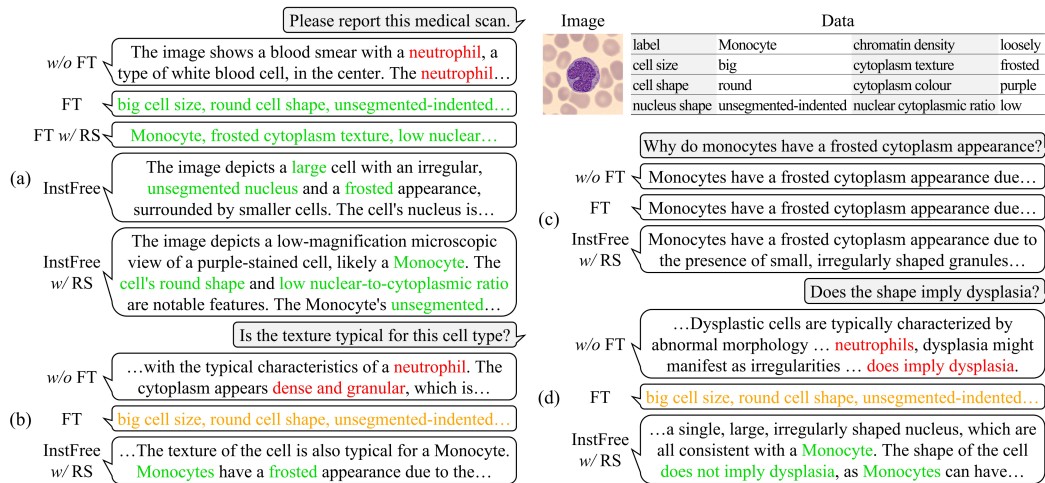

Figure 6: Qualitative VQA results on the WBCAtt dataset. Correct responses are marked in green, incorrect responses are marked in red, and responses that do not follow instructions are marked in orange.

tify images more precisely. InstFree and InstFree *w/* RS generate correct and natural text responses, whereas FT generates responses in the format of our medical reports. This suggests that using text instructions similar to those employed during fine-tuning may lead to overfitting (e.g., fixed-format outputs) or data leakage, highlighting the advantages of using the momentum proxy instruction in terms of generation quality and safety. Furthermore, response shuffling results in inconsistent output ordering in FT *w/* RS, whereas InstFree *w/* RS generates natural responses, demonstrating the greater robustness of our approach to output variability. In the case of attribute-related instruction (b), InstFree *w/* RS accurately identifies the cell type and generates responses based on the appropriate fine-tuned attributes, whereas *w/o* FT misclassifies it as a neutrophil and generates responses accordingly (e.g., dense and granular; the correct is frosted). Meanwhile, although the correct fine-tuned attributes are present in the response, FT fails to follow the instructions and generates responses that exhibit *image-dependence*. These results indicate that our method is highly sensitive to the given instructions and confirm its superior instruction-following capability. For instruction (c), which requires medical knowledge of relevant attributes, all models generate natural and accurate responses, demonstrating that they retain their pre-trained knowledge even after fine-tuning. In the case of challenging instruction (d), *w/o* FT identifies the cell as a neutrophil and implies dysplasia, whereas InstFree *w/* RS recognizes that the shape is commonly seen in monocytes and therefore does not imply dysplasia. Meanwhile, FT generates a response that does not follow the instruction, similar to (b). These results demonstrate that our approach enables the model to respond flexibly to diverse instructions, without requiring expert-level human effort in dataset construction. In summary, the qualitative VQA analysis not only supports the quantitative success but also demonstrates the strengths of the proposed method beyond accuracy.

# 6 CONCLUSION

We introduce an instruction-free tuning framework specifically tailored for the efficient fine-tuning of medical LVLMs. Our proposed momentum proxy instruction preserves the instruction-following capability of the pre-trained LVLM during fine-tuning, while simultaneously promoting refined updates to the visual encoder. This enables the fine-tuned model to flexibly respond to domain-specific instructions, even without explicit instructions during fine-tuning. Additionally, applying response shuffling during fine-tuning further mitigates the model's over-reliance on previous words, facilitating more effective fine-tuning. Experimental results demonstrate that our method achieves superior accuracy across the SKINCON, WBCAtt, and CBIS datasets, significantly enhancing fine-tuning efficiency in medical domains.

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

APPENDIX

## A ALGORITHM

The overall instruction-free tuning procedure with momentum proxy instruction is summarized in Algorithm A.1.

---

**Algorithm A.1** Instruction-free tuning process.

---

1: **Input:** Vision encoder $g$, language model $f$, ground truth output $y$, learning rate $\eta$, momentum coefficient $\alpha$
2: $t \leftarrow \mathcal{N}(0, \sigma^2)$
3: **while** not warmed-up **do**
4:   $y \leftarrow \text{RS}(y)$  // Response Shuffling
5:   **Compute** $\mathcal{L} \leftarrow L(g, t)$  // See Eq. 1
6:   $t \leftarrow t - \eta \nabla_t \mathcal{L}$
7: **end while**
8: $\bar{t} \leftarrow t$
9: **while** not converged **do**
10:   $\bar{t} \leftarrow \alpha \bar{t} + (1 - \alpha) t$
11:   $y \leftarrow \text{RS}(y)$
12:   **Compute** $\mathcal{L} \leftarrow L(g, \bar{t})$
13:   $g \leftarrow g - \eta \nabla_g \mathcal{L}; \quad t \leftarrow \bar{t} - \eta \nabla_{\bar{t}} \mathcal{L}$
14: **end while**

---

## B RANDOMIZED TEXT INSTRUCTIONS.

The list of randomized text instructions used for fine-tuning is summarized in Table A.1.

Table A.1: List of randomized text instructions.

---

"Analyze the image in a comprehensive and detailed manner.", "Analyze the image thoroughly and in detail.", "Break down the elements of the image in detail.", "Can you describe the image for me?", "Can you provide a brief summary of this radiograph?", "Can you provide a description of this medical scan?", "Can you provide a quick summary of this image?", "Can you provide a radiology report for this medical image?", "Can you provide a report summary for this medical scan?", "Can you summarize the images presented?", "Characterize the image with a well-detailed description.", "Clarify the contents of the displayed image in great detail.", "Could you provide a detailed description of what is shown in the picture?", "Create a compact narrative representing the image.", "Describe the composition and the subjects in this picture.", "Describe the following image in detail.", "Describe the image concisely.", "Describe the image in a detailed and informative manner.", "Describe the medical image you see.", "Describe the regions of interest in this scan.", "Describe the structures involved in this medical image.", "Describe this medical scan.", "Examine the image closely and share its details.", "Explain the various aspects of the image.", "Explain the visual content of the image.", "Give a detailed account of the given image.", "Give a short, clear explanation of the image.", "Give an elaborate explanation of the image you see.", "Narrate the contents of the image with precision.", "Offer a brief but comprehensive description of the image.", "Offer a succinct explanation of the image.", "Offer a thorough analysis of the image.", "Please caption this medical scan.", "Please describe this picture.", "Please generate a radiology report for this scan.", "Please provide a caption for this medical image.", "Please report this medical scan.", "Please write a radiology report for this image.", "Please write a radiology report for this scan.", "Present a concise description of the image's key features.", "Provide a rich, descriptive narrative of the image.", "Relay a brief, clear account of the image shown.", "Render a clear and concise summary of the image.", "Share a comprehensive rundown of the presented image.", "Share a concise interpretation of the image provided.", "Summarize the visual content of the image clearly.", "Walk through the important details of the image.", "What are the findings presented in this medical scan?", "What can you infer from this picture?", "What details stand out in this image?", "What is depicted in the image?", "What is depicted in this picture?", "What is the main focus of this photograph?", "What is the medical significance of this image?", "What is this photo about?", "Write a detailed description of the given image.", "Write a short but informative summary of the image.", "Write an exhaustive depiction of the given image."

---

## C  ADDITIONAL QUALITATIVE VQA RESULTS

Table A.2: Additional qualitative VQA results on the SKINCON dataset.

| | |
|---|---|
| Image | 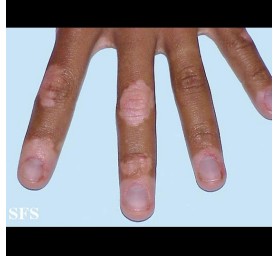 |
| Text | patch, white(hypopigmentation) |
| **Give a short explanation of the image.** | |
| *w/o* FT | The image shows a hand with fingers extended, set against a light blue background. The hand is positioned in the center of the image, with the fingers spread apart and the palm facing downwards. The fingernails are short and well-manicured, and the skin tone is a medium brown color. 

 In the bottom-left corner of the image, there is a watermark that reads "SFS" in small, white letters. The overall atmosphere of the image suggests that it may be a stock photo or a medical illustration, possibly used to demonstrate proper hand positioning or to highlight the appearance of the fingers and nails. |
| FT | patch, white(hypopigmentation) |
| InstFree *w/* RS | The image shows a hand with a white patch on the middle finger, which is a common symptom of vitiligo. |
| **What abnormal findings are observed?** | |
| *w/o* FT | The image shows a hand with a noticeable abnormality. The fingers are short and stubby, with a characteristic "mitten" appearance. This is due to the absence of the distal phalanges, which are the bones at the tips of the fingers. The proximal interphalangeal joints are also affected, resulting in a flexion deformity. The thumb is also affected, with a short and stubby appearance. 

 This condition is known as "mitten hand" or "lobster claw hand," and it is a rare congenital anomaly. It is characterized by the absence or hypoplasia of the distal phalanges, resulting in a mitten-like appearance of the hand. The condition can be caused by various genetic mutations or environmental factors during fetal development. |
| FT | patch, white(hypopigmentation) |
| InstFree *w/* RS | The image shows a white plaque on the dorsal surface of the middle finger. This is an example of a white patch, which is a type of flat, white lesion that can be seen on the skin. White patches are often caused by a lack of pigment in the skin, and they can be caused by a variety of factors, including genetics, sun exposure, and certain medical conditions. 
 Answer: White patch |

Table A.3: Additional qualitative VQA results on the SKINCON dataset.

| | |
|---|---|
| Image | 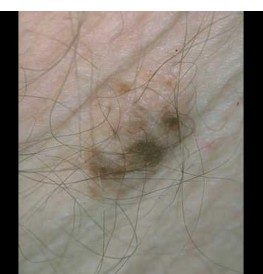 |
| Text | papule, brown(hyperpigmentation) |

**Give a short explanation of the image.**

*w/o* FT — The image presents a close-up view of a person's skin, specifically focusing on a mole or birthmark situated in the center of the frame. The mole is characterized by its dark brown color and irregular shape, featuring a slightly raised surface. It is surrounded by a scattering of light brown hairs, which are visible throughout the image.

The surrounding skin appears to be pale and smooth, with a subtle sheen that suggests it may be well-moisturized. The overall atmosphere of the image is one of intimacy and focus, as if the viewer is being invited to examine the mole up close. The use of a close-up shot and the emphasis on the mole's texture and color create a sense of immediacy and attention to detail, drawing the viewer's eye to the subject of the image.

FT — papule, brown(hyperpigmentation)

InstFree *w/* RS — The image depicts a brown papule on the skin, which is a small, raised area of skin that is typically brown in color. The papule is surrounded by a few strands of hair, and the surrounding skin appears to be pale. The image is likely a medical illustration or photograph used to aid in the diagnosis of a skin condition.

**What abnormal findings are observed?**

*w/o* FT — The image shows a close-up view of a person's skin, with a dark brown or black mole in the center. The mole is irregularly shaped and has a rough, uneven surface. It is surrounded by a few strands of hair.
* Mole:
- Location: Center of the image
- Color: Dark brown or black
- Shape: Irregular
- Surface: Rough, uneven
* Hair:
- Color: Light brown
- Length: Short
- Distribution: Scattered around the mole
The image suggests that the person may have a skin condition or a mole that needs to be examined by a dermatologist.

FT — papule, brown(hyperpigmentation)

InstFree *w/* RS — The image shows a brown papule with a brown papule. The brown papule is a small, flat, brown-colored growth on the skin. It is typically less than 1 cm in diameter and can be found anywhere on the body. The brown papule is a common benign skin lesion that can be caused by a variety of factors, including sun exposure, genetics, and trauma. It is usually asymptomatic and does not require treatment. However, if the brown papule becomes larger or changes in color, it may be a sign of a more serious condition and should be evaluated by a healthcare professional.

Table A.4: Additional qualitative VQA results on the WBCAtt dataset.

| | |
|---|---|
| Image |  |
| Text | small cell size, round cell shape, segmented-multilobed nucleus shape, low nuclear cytoplasmic ratio, densely chromatin density, clear cytoplasm texture, light blue cytoplasm colour, granularity, coarse granule type, purple granule colour, Basophil |

**Give a short explanation of the image.**

| | |
|---|---|
| *w/o* FT | The image shows a blood smear with a purple-stained cell in the center, surrounded by other cells. The cell in the center is likely a neutrophil, which is a type of white blood cell that plays a crucial role in the immune system.
Key Features:
* Cell in the Center: The cell in the center of the image is stained purple and has a distinctive morphology, with a multilobed nucleus and numerous granules in the cytoplasm. This suggests that it is a neutrophil, which is a type of white blood cell that plays a crucial role in the immune system.
* Other Cells: The other cells in the image are also stained purple, but they do not have the same morphology as the cell in the center. They appear to be red blood cells, which are responsible for carrying oxygen throughout the body.
* Background: The background of the image is a light-colored substance that appears to be a type of medium or agar. |
| FT | small cell size, round cell shape, irregular nucleus shape, low nuclear cytoplasmic ratio, densely chromatin density, clear cytoplasm texture, light blue cytoplasm colour, granularity, coarse granule type, purple granule colour, Basophil |
| InstFree *w/* RS | The image shows a purple, round cell with a coarse, irregular nucleus and clear cytoplasm. The cell is densely packed with small, light-colored granules. The background of the image is a light yellow color with several other cells visible, including red blood cells and white blood cells. The overall appearance of the image suggests that it is a microscopic view of a blood sample, possibly taken for diagnostic purposes. |

**Is the texture typical for this cell type?**

| | |
|---|---|
| *w/o* FT | Step 1: Identify the cell type.
The image shows a blood smear with a cell in the center that has a large, darkly staining nucleus and a small amount of cytoplasm. The cell is surrounded by red blood cells.
Step 2: Determine the texture of the cell.
The cell has a coarse, granular texture, which is typical for this cell type.
Step 3: Conclusion.
The texture of the cell is typical for a neutrophil, which is a type of white blood cell that plays a key role in the immune system.
Answer: Yes |
| FT | small cell size, round cell shape, irregular nucleus shape, low nuclear cytoplasmic ratio, densely chromatin density, clear cytoplasm texture, light blue cytoplasm colour, granularity, coarse granule type, purple granule colour, Basophil |
| InstFree *w/* RS | The cell in the image is a Basophil, which is a type of white blood cell. Basophils are characterized by their small size and round shape, with a clear cytoplasm and a purple-stained nucleus. The texture of the cell is typically coarse and irregular, with a granular appearance due to the presence of granules in the cytoplasm. The granules are densely packed and give the cell a distinctive purple color when stained with certain dyes.
Answer: Yes |

Table A.5: Additional qualitative VQA results on the WBCAtt dataset.

| | |
|---|---|
| Image |  |
| Text | big cell size, irregular cell shape, unsegmented-indented nucleus shape, low nuclear cytoplasmic ratio, loosely chromatin density, cytoplasm vacuole, frosted cytoplasm texture, blue cytoplasm colour, Monocyte |
| **Give a short explanation of the image.** | |
| *w/o* FT | The image shows a blood smear, which is a diagnostic tool used in hematology to examine the morphology of blood cells. The smear is typically prepared by spreading a small amount of blood onto a glass slide and allowing it to dry. The resulting image reveals various types of cells, including red blood cells, white blood cells, and platelets. Key Features: * Red Blood Cells (RBCs): These cells are typically round or oval in shape and have a central pallor. They are responsible for carrying oxygen from the lungs to the body's tissues. * White Blood Cells (WBCs): These cells are larger and more irregularly shaped than RBCs. They play a crucial role in the immune system, helping to fight infections and diseases. * Platelets: These small, irregularly shaped cells are involved in blood clotting and wound healing. Importance of Blood Smears: Blood smears are an essential diagnostic tool in hematology... |
| FT | big cell size, irregular cell shape, irregular nucleus shape, low nuclear cytoplasmic ratio, loosely chromatin density, cytoplasm vacuole, frosted cytoplasm texture, blue cytoplasm colour, Monocyte |
| InstFree *w/* RS | The image depicts a Monocyte, a type of white blood cell, characterized by its irregular shape and low nuclear-to-cytoplasmic ratio. The cytoplasm is blue and contains clear cytoplasmic vacuoles, while the nucleus is irregularly shaped and densely stained. The background of the image shows red blood cells, which are round and pink in color. |
| **Is the texture typical for this cell type?** | |
| *w/o* FT | Step 1: Identify the cell type. The image shows a neutrophil, which is a type of white blood cell. Step 2: Determine the texture of the cell. Neutrophils have a characteristic texture that is typically described as "toxic" or "dysplastic." This means that the cell's cytoplasm appears abnormal and may be more dense or granular than normal. Step 3: Compare the texture to typical characteristics. The texture of the cell in the image appears to be consistent with the typical characteristics of a neutrophil. The cytoplasm is dense and granular, which is consistent with the expected appearance of a mature neutrophil. Answer: Yes |
| FT | big cell size, irregular cell shape, irregular nucleus shape, low nuclear cytoplasmic ratio, loosely chromatin density, cytoplasm vacuole, frosted cytoplasm texture, purple blue cytoplasm colour, Monocyte |
| InstFree *w/* RS | The cell in the image is a Monocyte, which is a type of white blood cell. Monocytes are characterized by their irregular shape and low nuclear-to-cytoplasmic ratio. They have a light blue cytoplasm with a frosted appearance due to the presence of numerous small cytoplasmic vacuoles. The nucleus is irregularly shaped and has a coarse chromatin texture. The texture of the cytoplasm is typically low and frosted, which is a characteristic feature of monocytes. This is because monocytes have a high content of cytoplasmic organelles, such as lysosomes and mitochondria, which give the cytoplasm a frosted appearance under the microscope. Therefore, the texture of the cytoplasm in the image is typical for a Monocyte. |

Table A.6: Additional qualitative VQA results on the CBIS dataset.

| | |
|---|---|
| Image | 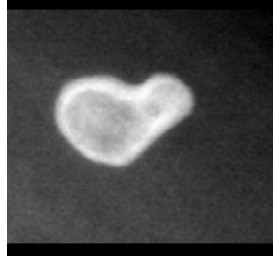 |
| Text | lucent center calc type, calcification, benign |

**Give a short explanation of the image.**

| | |
|---|---|
| *w/o* FT | The image depicts a white, irregularly shaped object against a dark background. The object is positioned in the center of the image, with its top-right side slightly closer to the viewer than its bottom-left side. It has a rounded, heart-like shape, with a small, rounded protrusion on its right side. The object is surrounded by a bright white glow, which is more pronounced on its left side. The background of the image is a dark gray color, with a subtle texture that resembles a grainy or speckled pattern. Overall, the image appears to be a close-up photograph of a small, white object, possibly a piece of debris or a fragment of a larger item. |
| FT | coarse-round and regular-lucent center calc type, calcification, benign |
| InstFree *w/* RS | The image depicts a calcification of a benign type, specifically a lucent center calcification with a benign-appearing rim. This type of calcification is commonly associated with benign conditions, such as benign calcification of the breast tissue. The lucent center calcification is a characteristic feature of this type of calcification, which is often seen in benign conditions. The benign-appearing rim surrounding the calcification is also a key feature of this type of calcification, which helps to distinguish it from malignant calcifications. Overall, the image suggests that the calcification is likely benign in nature. |

**What abnormal findings are observed?**

| | |
|---|---|
| *w/o* FT | Step 1: Identify the abnormal findings.
The image shows a radiograph of a bone, which appears to be a femur. The abnormal finding is the presence of a fracture in the femur.
Step 2: Describe the abnormal findings.
The fracture is located in the proximal femur, near the hip joint. It appears to be a transverse fracture, meaning it runs perpendicular to the long axis of the bone.
Step 3: Provide additional information about the abnormal findings.
The fracture is likely due to trauma, such as a fall or car accident. The patient may experience pain, swelling, and limited mobility in the affected limb.
Step 4: Summarize the abnormal findings.
The abnormal finding in this radiograph is a transverse fracture in the proximal femur, likely caused by trauma. |
| FT | coarse calc type, calcification, benign |
| InstFree *w/* RS | The image shows a calcification in the shape of a benign lucent center with a benign calcification rim. |

Table A.7: Additional qualitative VQA results on the CBIS dataset.

| | |
|---|---|
| Image | 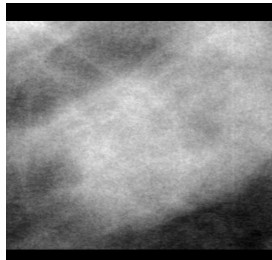 |
| Text | oval mass shape, obscured mass margins, mass, benign |

**Give a short explanation of the image.**

| | |
|---|---|
| *w/o* FT | The image appears to be a black and white photograph of a textured surface, possibly a piece of fabric or paper. The surface is predominantly white, with darker areas scattered throughout, creating a mottled effect. The texture of the surface is visible, with subtle variations in tone and shading that give it a tactile quality. The overall effect is one of softness and subtlety, with the image inviting the viewer to touch or explore the surface further. |
| FT | oval mass shape, obscured mass margins, mass, benign |
| InstFree *w/* RS | The image depicts a mass on a mammogram, characterized by a round, oval, or lobulated mass with obscured or obscured mass margins. The mass is benign, as indicated by its oval shape and obscured mass margins. The mass is located in the upper outer quadrant of the breast, and its size is not specified. The mass is not described as calcified, mass-like, or obscured mass margins, which are common characteristics of benign masses. The mass is also not described as mass-like, which is a characteristic of malignant masses. Overall, the mass is benign and does not appear to be cancerous. |

**Are the visual characteristics typical for this diagnosis?**

| | |
|---|---|
| *w/o* FT | The image provided appears to be a mammogram, which is a type of medical imaging used to examine the breast tissue. The image shows a dense, white area with some darker areas scattered throughout. The density of the image suggests that it may be a mammogram, as mammograms are typically taken in a dense, white format. |
| | However, without more information or context, it is difficult to say for certain whether the visual characteristics of the image are typical for this diagnosis. Mammograms can show a wide range of abnormalities, including tumors, cysts, and calcifications, so it's possible that the image could be showing something abnormal. On the other hand, it's also possible that the image is normal and just shows the natural variation in breast tissue. |
| | To determine whether the visual characteristics of the image are typical for this diagnosis, it would be helpful to have more information about the patient's medical history, the results of any other diagnostic tests, and the interpretation of the mammogram by... |
| FT | oval mass shape, obscured mass margins, mass, benign |
| InstFree *w/* RS | To determine if the visual characteristics are typical for this diagnosis, we need to identify the diagnosis first. Based on the description of a mass with obscured mass margins and oval mass shape, it is likely that the diagnosis is a benign mass, possibly a fibroadenoma. |
| | Benign Mass Characteristics: |
| | - Oval or round shape |
| | - Obscured or ill-defined mass margins |
| | - Benign mass |
| | Conclusion: |
| | Yes, the visual characteristics are typical for this diagnosis. |

