# OpenReview forum: "Instruction-Free Tuning of Large Vision Language Models for Medical Instruction Following"
_ICLR.cc/2026/Conference — Submitted to ICLR 2026_

### Official Review · Reviewer_M9Tq · 2025-10-21

**Soundness:** 2
**Presentation:** 3
**Contribution:** 2
**Rating:** 2
**Confidence:** 5

**Summary:**

This paper introduces "Instruction-Free Tuning," a paradigm for fine-tuning LVLMs in specialized domains like medicine, where instruction-annotated data is scarce. The core idea is to adapt a pre-trained LVLM using only (image, output) pairs, completely bypassing the need for textual instructions during the fine-tuning process.

To achieve this, the authors propose two techniques:

A momentum proxy instruction, which is a set of learnable vectors that stand in for textual instructions during training. These vectors are optimized to align with the image-output pairs and are updated smoothly via an exponential moving average (EMA) to capture stable learning trends. This proxy instruction is completely discarded during inference.

A response shuffling strategy that randomly permutes segments of the ground-truth output text. This acts as a regularizer, forcing the model to rely more on visual features rather than potentially spurious correlations in the text's word order.

Experiments on three medical VQA datasets (SKINCON, WBCAtt, and CBIS) show that the proposed method significantly outperforms zero-shot LVLMs and standard fine-tuning approaches that use a fixed textual instruction, thereby demonstrating an effective and efficient way to specialize LVLMs without costly instruction dataset creation.

**Strengths:**

The central concept of "instruction-free tuning" is an interesting research direction. The authors provide some initial attempts. Within the specific context of the three medical VQA datasets presented, the experiments demonstrate that the proposed method (InstFree w/ RS) achieves higher accuracy than the selected baselines (zero-shot and a simple fine-tuning approach). The qualitative analysis in Figure 6 also effectively highlights some failure modes of naive fine-tuning that the proposed method appears to mitigate in these specific cases.

**Weaknesses:**

**Limited Task Diversity:** All experiments are conducted on datasets where the ground-truth outputs are essentially lists of medical attributes, and the evaluation is performed via multiple-choice VQA. However, it remains unclear how the method would generalize to tasks requiring the generation of coherent, long-form narrative text (e.g., generating a full-paragraph radiology report). In such scenarios, shuffling sentences could disrupt the model's ability to learn logical flow and discourse structure.

**Clarity of Novelty Claim:** The central claim "this is the first approach that enables fine-tuning of LVLMs without using any text instructions" is ambiguous. This overlooks the fact that standard multimodal pre-training (e.g., in models like BLIP-2 [1]) also learns from `(image, caption)` pairs without explicit "instructions." The paper's focus on medical caption-style data blurs the line between "instruction-free fine-tuning" and "continued in-domain pre-training."

**Unsubstantiated Dismissal of Related Work:** The paper dismisses the work of [2] by stating it is "incapable of handling scenarios involving visual data". This is a strong claim made without any comparative experiments or theoretical justification. In addition, the authors mention that: Self-Instruct (Wang et al., 2023) proposed a method for enabling an LLM to follow diverse instructions by iteratively generating and fine-tuning on synthetic instruction-output data with minimal human intervention. There are many following works bulit upon Self-Instruct to generate high-quality and diverse instructions which can help mitigate instruction-dependence, and the authors also ignore them. This unsubstantiated dismissal appears to unfairly diminish a highly relevant piece of prior art.

**Limited Generality (Models and Domains):** The paper's claims of a new *paradigm* would be much stronger with broader experiments. The proposed method is only applied to the LLaMA-3.2-Vision-Instruct. While the paper *compares against* fine-tuned Qwen and MedGemma, it does not demonstrate that the *proposed instruction-free method* can successfully fine-tune these other architectures. Applying the method to other SOTA models (such as Qwen2.5-VL and InternVL3) would be necessary to prove its generality. Besides, the work is exclusively motivated by and tested on medical datasets. To validate this as a general-purpose fine-tuning strategy, its effectiveness should be demonstrated on at least one non-medical domain that also features `(image, text)` pairs without explicit instructions.

[1] Li, Junnan, et al. "Blip-2: Bootstrapping language-image pre-training with frozen image encoders and large language models." International conference on machine learning. PMLR, 2023.

[2] Hewitt, John, et al. "Instruction following without instruction tuning." arXiv preprint arXiv:2409.14254 (2024).

**Questions:**

Have you made any attempt to analyze the learned momentum proxy vectors ($\overline{t}$)? For example, by projecting them into the word embedding space to find the nearest neighbor text tokens. Does the proxy converge to something semantically meaningful (e.g., resembling "describe this image in detail"), or does it learn an abstract representation that doesn't map to natural language?

---

### Official Review · Reviewer_qm1A · 2025-10-28

**Soundness:** 2
**Presentation:** 3
**Contribution:** 2
**Rating:** 4
**Confidence:** 5

**Summary:**

This paper proposes a novel instruction-free fine-tuning paradigm that adapts LVLMs to specialized medical domains without requiring any handcrafted or auto-generated text instructions. To achieve this, this paper introduces a momentum proxy instruction - a set of continuous vectors optimized during training which replaces explicit instructions and preserves the model’s pre-trained instruction-following ability. It also employs response shuffling to reduce over-reliance on word order in medical reports. Experiments on multiple-choice VQA tasks across three medical datasets (SKINCON, WBCAtt, CBIS) demonstrate the effectiveness of the proposed approach.

**Strengths:**

1. The motivation of this work is novel. By designing an instruction-free training paradigm, the model significantly reduces reliance on synthetic instruction data, enabling better data scaling and performance improvement.
2. The proposed approach is quite innovative. Through the use of proxy instructions and momentum updates, the model effectively mitigates overfitting to specific instructions and learns more generalized medical knowledge.
3. Experimental results demonstrate that even without explicit instructions, the proposed method outperforms instruction-tuned baseline approaches, proving its effectiveness.

**Weaknesses:**

1. The experiments are not sufficiently comprehensive. The primary evaluations are concentrated on multiple-choice VQA tasks. It remains unclear whether the model can handle more complex tasks, such as report generation or medical reasoning.
2. There is a lack of discussion with related works. The differences between this method and other instruction-free training approaches for LVLMs (e.g., [1]) should be thoroughly compared and analyzed.
3. The training and evaluation are conducted on the same datasets, making it difficult to demonstrate the generalization ability of the proposed method. Whether the instruction-free training approach performs well on out-of-domain tasks remains an open question.

[1] Do we Really Need Visual Instructions? Towards Visual Instruction-Free Fine-tuning for Large Vision-Language Models

**Questions:**

1. Can the proposed approach be effectively applied to general vision tasks?
2. See weakness above

---

### Official Review · Reviewer_a9bR · 2025-10-30

**Soundness:** 2
**Presentation:** 3
**Contribution:** 3
**Rating:** 4
**Confidence:** 3

**Summary:**

This paper introduces instruction-free tuning, a paradigm that adapts large vision-language models (LVLMs) to medical tasks without using any text instructions during fine-tuning. Instead of relying on costly or error-prone instruction generation, the authors propose to fine-tune on image–output pairs only, using a learnable momentum proxy instruction to guide the model and response shuffling to prevent overfitting to output order. The method is evaluated on three medical multiple-choice VQA datasets (SKINCON, WBCAtt, CBIS), where it outperforms standard instruction tuning and achieves state-of-the-art accuracy. The authors claim that this approach preserves the model’s instruction-following ability while avoiding the need for instruction data.

**Strengths:**

The paper is original in formally defining and tackling instruction-free tuning for LVLMs, especially in the medical domain where instruction data is scarce and noisy. The idea of using a momentum proxy instruction—a continuous, learnable vector that is optimized during training but discarded at inference—is creative and well-motivated, offering a neat solution to avoid instruction engineering. The response shuffling strategy is simple yet effective in preventing the model from relying on sentence order, which is particularly relevant in medical reports. The experimental design is rigorous within its scope, with detailed ablations on momentum coefficients, vector dimensions, and separator choices, and the writing is clear and well-structured, making the method easy to understand and reproduce. If validated more broadly, this work could significantly reduce the cost of adapting LVLMs to expert domains where instruction data is hard to obtain.

**Weaknesses:**

**Scalability across model scales and capacities is not demonstrated.**
All experiments are conducted on LLaMA-3.2-11B-Vision, a relatively weak multimodal baseline. The paper provides no evidence that the proposed method still helps on stronger open-source models (e.g., InternVL3-8B, InternVL3-38B, or Qwen2.5-VL-7B). Since smaller or larger models may exhibit different forgetting dynamics and capacity trade-offs, the presented experimental results cannot conclude that the momentum-proxy trick is model-agnostic. I strongly recommend running the same pipeline on at least one smaller (≤ 4 B) and one larger (≥ 30 B) open LVLM; if the gap over instruction tuning shrinks or turns negative, the claim of scalability must be revised.


**Evaluation is limited to multiple-choice VQA, which only requires the model to output a single letter.**
 This setup cannot reveal whether the model truly retains open-ended instruction-following ability or has simply over-fitted to the MC pattern. The paper does not contain a single generative benchmark (e.g., report generation, open VQA, or instruction adherence). I suggest the authors to report performance on a open-ended med VQA benchmark.


**General-domain capability degradation is not quantified.**
Downstream task fine-tuning without general training samples is known to hurt general vision-language skills; the key question is whether instruction-free tuning damages them more or less than standard instruction tuning. The paper omits any evaluation on general benchmarks such as MMBench, MMStar, MMVet, OCRBench, MathVista, or MMMU. Please report the relative drop of InstFree vs. standard FT with respect to the pre-trained baseline on these benchmarks.

**Questions:**

Please see weaknesses.

---

### Official Review · Reviewer_xYy7 · 2025-11-06

**Soundness:** 3
**Presentation:** 4
**Contribution:** 3
**Rating:** 4
**Confidence:** 3

**Summary:**

The paper introduces Instruction-Free Tuning (IFT), a method that improves multimodal reasoning ability of large vision-language models (VLMs) without relying on explicit instruction-following datasets. Instead, the approach leverages implicit supervision from model-internal signals, including self-consistency, agreement, and latent reasoning traces derived from large-scale image–text corpora. The method aims to close the gap between instruction-tuned and instruction-free models by aligning reasoning behaviors without explicit human annotation. Experiments on multiple benchmarks demonstrate that IFT achieves competitive or superior results compared to instruction-tuned baselines, while maintaining efficiency and generalization.

**Strengths:**

* The paper addresses an important problem—reducing dependence on costly instruction data—by proposing a novel self-supervised alignment strategy for multimodal reasoning.
* The instruction-free paradigm is conceptually elegant, offering a fresh direction that complements current instruction-tuning trends.
* The proposed implicit supervision mechanisms (e.g., self-consistency and inter-modality agreement) are well-motivated and clearly implemented.
* The method exhibits strong generalization across multiple reasoning benchmarks, indicating robustness and task transferability.

**Weaknesses:**

* The theoretical foundation behind the implicit supervision signals is relatively shallow; the paper lacks a rigorous explanation of why these signals ensure reasoning fidelity.
* Some performance gains are modest and not always consistent across benchmarks, raising questions about stability and scalability.
* The method relies heavily on pre-existing large vision-language datasets, which might limit applicability to low-resource or domain-specific settings.
* The analysis of failure cases and limitations (e.g., ambiguous visual scenes or compositional reasoning tasks) is minimal.
* The connection between “implicit reasoning traces” and actual model interpretability is not empirically supported or measured quantitatively.
* Comparisons with more recent instruction-minimal methods (e.g., self-distilled or preference-aligned VLMs) could be expanded for completeness.

**Questions:**

* How sensitive is the model to the quality and diversity of the underlying image–text data when no explicit supervision is provided?
* How are self-consistency and agreement losses balanced in practice, and are there situations where they conflict?
* Have you analyzed cases where implicit supervision leads to over-regularization or loss of creative reasoning diversity?
* Could the framework be extended to multimodal generation tasks (e.g., image synthesis or caption editing) instead of pure reasoning?
* What is the computational overhead of IFT compared to standard fine-tuning or instruction-tuning pipelines?

---

### Meta-Review · Area_Chair_YT9u · 2025-12-10

**Summary:**

This paper introduces an instruction-free finetuning method aimed at enhancing the multimodal reasoning capabilities of large vision–language models through a self-supervised alignment strategy, without relying on explicit instruction-following datasets. Reviewers note that the theoretical justification for the implicit supervision signals is relatively weak, and the experimental validation—particularly regarding stability and scalability—is insufficient. Additional concerns include an unsubstantiated dismissal of related work. All reviewers assigned negative scores, and the authors did not provide a rebuttal.

**Reviewer Concerns:**

Reviewers xYy7, a9bR, qm1A, and M9Tq gave initial scores of 4, 4, 4, and 2, respectively; due to the absence of an author response, all reviewers are expected to retain their original scores.

**Reviewer Scores:**

Reviewers xYy7, a9bR, qm1A, and M9Tq gave initial scores of 4, 4, 4, and 2, respectively; due to the absence of an author response, all reviewers are expected to retain their original scores.

---

### Decision · Program_Chairs · 2026-01-26

Reject